# Can a Mixture of Farnesene Isomers Avert the Infestation of Aphids in Sugar Beet Crops?

**DOI:** 10.3390/insects15100736

**Published:** 2024-09-24

**Authors:** Denise Kuhn, Nils Nägele, Till Tolasch, Georg Petschenka, Johannes L. M. Steidle

**Affiliations:** 1Department of Chemical Ecology 190t, Institute of Biology, University of Hohenheim, 70599 Stuttgart, Germany; tolasch@uni-hohenheim.de; 2Department of Applied Entomology 360c, Institute of Phytomedicine, University of Hohenheim, 70599 Stuttgart, Germany; nils.naegele@uni-hohenheim.de (N.N.); georg.petschenka@uni-hohenheim.de (G.P.); 3KomBioTa—Center of Biodiversity and Integrative Taxonomy, University of Hohenheim, 70599 Stuttgart, Germany

**Keywords:** pest control, insecticide replacement, (E)-ß-farnesene, alarm pheromone, aphids, sugar beet

## Abstract

**Simple Summary:**

Many studies have shown that pesticide use has negative environmental effects. Therefore, alternative measures to manage insect pests and protect plant health are needed. Potential tools include chemicals that cause behavioral changes in pests and are already a part of the integrated pest management toolbox. Aphids are a significant pest in sugar beet (*Beta vulgaris*). They cause visual damage and act as vectors for various phytopathogens that reduce the yield. When attacked by predators, aphids emit an alarm pheromone, which causes other aphids to stop feeding, move away, and sometimes even abandon the host plant. We tested whether the artificially release of this pheromone could be an environmentally friendly alternative to prevent aphid infestation in sugar beet. Application of the pheromone resulted in reduced aphid densities in experimental sugar beet patches at two out of three experimental sites. This indicates the potential of this pheromone as an aphid repellent in sugar beet production. However, more research is required to ensure the reliability of this method.

**Abstract:**

The negative impact of pesticides on the environment and the potential of pest species to develop pesticide resistance make it necessary to explore new methods of pest control. Pheromones and other behavior-modifying semiochemicals are already important in integrated pest management (IPM). (E)-ß-farnesene (EBF) is a semiochemical that acts as an alarm pheromone in aphids. Upon perception of EBF, aphids stop feeding, move away, and sometimes even abandon the host plant. The aphids *Myzus persicae* and *Aphis fabae* are significant crop pests and vectors of many harmful phytopathogens affecting sugar beet (*Beta vulgaris*). Field trials were conducted at different locations in Germany to test whether dispensers containing a mixture of farnesene isomers (FIMs) including EBF were able to reduce the infestation of these species on sugar beet. Our results showed a reduction in aphid abundance in the FIM-treated patches in two out of three sites. Therefore, we hypothesize that FIM dispensers could prevent aphid infestation and could be used in combination with other IPM measures. However, more research is required to increase the effect and ensure the reliability of this method.

## 1. Introduction

There is general agreement that intensified agriculture contributes considerably to the decline in insect diversity and abundance [1,2]. Insecticides are very likely to play an important role in this process, affecting not only insect pests, but also non-target insects such as pollinators and predators [3,4,5,6,7]. Thus, there is a pressing need for alternative methods to replace or reduce insecticides. Synthetic semiochemicals have been found to be effective tools for pest control in various agricultural fields [8], especially in viticulture, fruit cultivation, forestry, and horticulture [9]. As natural compounds, they mediate communication between individuals within the same species (pheromones [10]) or of different species (allelochemicals [11]). Pheromones possess the capability of inducing long-lasting physiological or behavioral reactions in conspecifics [10], such as sexual attraction, mating, dispersion, oviposition, aggregation, and alarm behavior [10,12,13]. In pest management, synthetic pheromones are used mainly for (a) mating disruption, (b) monitoring, and (c) mass trapping (attract and kill). For instance, two important tortricid pests, *Eupoecilia ambiguella* (Hübner, 1796) (Lepidoptera: Tortricidae) in vineyards and *Cydia pomonella* (L., 1758) (Lepidoptera: Tortricidae) in orchards, are mainly controlled with pheromones [12]. Regarding allelochemicals, various studies have focused on the use of synthetic volatile secondary plant metabolites that are released by infested plants to defend themselves against herbivores, such as methyl salicylate (MeSA), (Z)-3-hexenol, dodecanoic acid (DDA), and (E)-ß-farnesene (EBF), as part of integrated pest management (IPM) [13,14,15]. These volatiles act either as repellents of pest insects [16], or to attract natural enemies to improve biological pest control [17], or in a ‘push and pull’ strategy, where pests (i.e., aphids) are lured away from the crop by repellents or deterrents (push) to establish themselves elsewhere where attractive stimuli are provided (pull) [14].

Many aphids use the sesquiterpene (E)-ß-farnesene (EBF) as an alarm pheromone, which they emit through their cornicles in the presence of natural enemies [18,19,20,21]. Upon detection, it causes short-term behavioral changes in other individual aphids. They stop feeding on their host plant or even withdraw their stylet from the plant tissue and move away from the volatile source, which could include dropping off the host plant [22,23,24,25]. Moreover, prolonged exposure to high levels of EBF can lead to the production of winged aphids in the next generation that are able to migrate from the “danger zone” to colonize another host plant [26,27,28]. Also, the normal population growth rate is disrupted and the rate of infestation is reduced [29,30]. In addition, under laboratory conditions, EBF has been shown to serve as a foraging kairomone [31] and attracts various natural enemies of aphids, such as ladybirds [31,32], syrphid flies [33], and parasitoid wasps [34]. Results from field experiments on EBF as an attractant for natural enemies to control aphids, however, are inconsistent [34,35,36,37]. Some studies have demonstrated the effect of EBF as a kairomone on various natural enemies under field conditions [34], but others showed that the observed effect was due to the exceptionally high concentrations of pheromone used [36].

Sugar beet *Beta vulgaris* subsp. *vulgaris* (Amaranthaceae) is mainly infested by two aphid species [38], *Aphis fabae* (Scopoli 1763) (Hemiptera: Aphididae) and *Myzus persicae* (Sulzer 1776) (Sternorrhyncha: Aphididae). While sucking the phloem, these aphids deprive the plants of important nutrients, resulting in retarded growth of sugar beet, loss of sugar content up to 50%, water stress, and leaf deformities [39,40]. In addition, both aphid species act as vector of a variety of viruses such as beet chlorosis virus (BCHV), beet mosaic virus (BtMV), and beet yellow virus (BYV). While the first two viruses induce orange-yellow hues on the surface of the leaves [41], BYV causes leaf blade discoloration with necrosis in the form of dots or dashes. BYV is the most prevalent virus of sugar beet in Central Europe, and frequently results in significant reductions in yield and sugar content [42].

In this study, we investigated the potential use of a commercially available mixture of farnesene isomers (FIMs) to control aphids. The mixture contained low levels of the aphid alarm pheromone EBF and was therefore suitable for use as a preventative method to avoid aphid infestation of sugar beets and virus transmission.

## 2. Materials and Methods

### 2.1. Experimental Sites

Experiments were performed on conventionally managed *B. vulgaris* fields in Mannheim (49°35′05.7″ N 8°27′18.0″ E), Heidelberg (49°22′25.8″ N 8°39′39.6″ E), and Dettenheim (49°10′59.2″ N 8°24′07.2″ E) (Southern Germany). No insecticides were applied to the crop during the experimental period. Sugar beet is usually sown at 110.000 seeds/ha, with about 80.000 seeds/ha emerging, giving 8 plants/m^2^.

### 2.2. Dispenser Preparation and Application

We used Eppendorf tubes (1.5 mL) as FIM dispensers, which were filled with 0.7 mL of a commercially available farnesene isomer mixture (FIM, Sigma Aldrich, St. Louis, MO, USA, product number: W383902), containing 15.1% EBF as well as 0.1% alpha-tocopherol as stabilizer. The mixture was used instead of pure EBF because it was more cost effective. We applied dispensers filled with FIMs onto conventionally managed *B. vulgaris* fields in three locations in Southern Germany and monitored the numbers of aphids on the plants. Immediately before application, the tubes were perforated with a needle four times just above the fill level (diameter of the perforations: 0.1 mm) to facilitate evaporation. FIM dispensers were placed in the field in spring as soon as sugar beets had emerged, and the first winged female aphids were detected in yellow pan traps. Dispensers were installed on a plastic stick 5 cm above the ground downwind (i.e., away from the control patches to avoid potential contamination, see arrows in Figure 1A,C,E) and checked weekly for potential drying out.

### 2.3. Sampling

The first sampling took place before FIM dispensers were placed in the field (labeled “0 days” in Figure 1A,C,E), to determine the initial population density of aphids. After installation of the dispensers, weekly samples were taken for seven consecutive weeks (i.e., 7, 14, 21, 28, 35, and 42 days after application). Sampling was performed by cutting all leaf material from randomly selected beet plants within each respective patch (see below) directly above the soil and preserving it in a Whirl-Pak^TM^ (125 mm × 380 mm, volume: 1065 mL, Sigma Aldrich, St. Louis, MO, USA) filled with 70% ethanol. Care was taken to ensure that the leaf material was completely covered with ethanol.

### 2.4. Experimental Setups

Different experimental designs were used at the three sites. The fact that no insecticide was allowed to be applied to these patches for the duration of the trial limited the areas that the farmers could make available for the trials. Therefore, the numbers and distribution of the dispensers needed to be adapted to the available patches. As a result, different sizes and shapes of patches were available. In addition, sowing times could not be standardized, due to the responsibilities of the individual farmers and varying weather conditions. The experimental and the control patches were not separated in the field.

In Mannheim, *B. vulgaris* was sown at the beginning of March 2023. Within the cultivated field, six patches in a row (W × L: 18 m × 50 m, 900 m^2^ each) were selected and staked (Figure 1A). The patches alternated between experimental patches with dispensers and control patches without dispensers. In each experimental patch, two FIM dispensers were placed directly at the edge, 30 m apart and 3 m from each end of the patch. During the experiment, 4 out of 7200 beet plants in each patch were sampled 7, 14, 21, 28, 35, and 42 days after FIM application. given that the sampled patches were smaller than the other patches (i.e., *n* = 12 for each treatment, 84 beet plants in total).

In Heidelberg, *B. vulgaris* was sown in April 2023 in a field measuring 120 m × 70 m (8400 m^2^). The field was divided into two equal patches of 120 m × 35 m (4200 m^2^) each (Figure 1C). In the experimental patches, five FIM dispensers were evenly spaced 20 m from each end of the field, 30 m apart horizontally, 15 m apart vertically. At each sampling time, 10 out of 33.600 beet plants per patch were randomly collected 7, 14, 21, 28, 35, and 42 days after FIM application.

In Dettenheim, we used a similar design as in Heidelberg. *B. vulgaris* was sown at the end of March 2023 in a field measuring 260 m × 150 m (39.000 m^2^). An area at the edge of the field, measuring 40 m × 90 m (3600 m^2^), was selected and divided into two equal patches (45 m × 40 m, each 1800 m^2^) (Figure 1E). Four FIM dispensers were evenly spaced in the experimental patch (distance between dispensers: 20 m vertical, 15 m horizontal). At each sampling time, 10 out of 14.400 beets per patch were randomly collected 7, 14, 21, 28, 35, and 42 days after FIM application.

### 2.5. Preparation of Samples

To obtain all arthropods from the beet leaves in the Whirl-Paks™ (Thermo Fisher Scientific, Waltham, MA, USA), the ethanol was poured through a sieve (mesh size 0.5 mm) and the sample bag was rinsed using ethanol. Then, the insects were counted and identified to species level using a stereomicroscope (VWR^®^ VisiScope^®^ 360 stereo zoom microscope, Avantor^®^, Ismaning, Germany). We aimed to differentiate the found aphids into “alates” and “nymphs”. However, the number of alates on the plants was sometimes too small to make a statistical analysis. So, we combined nymphs and alates.

### 2.6. Statistical Analyses

Data were analyzed using R software (version 4.0.3) [43,44] with pre-installed packages as well as the packages multcomp [45], car [46], and glmm [47]. Significance was assumed at *p* < 0.05, a tendency was assumed at 0.05 < *p* > 0.06. In the Heidelberg dataset, outliers were identified based on the four-sigma rule [48] and excluded from the dataset. An analysis of their influence revealed that the results did not depend on the presence of the outliers.

For comparing initial aphid numbers between the control and experimental patches (i.e., before the dispenser application took place, labeled as “0”), Wilcox rank tests were conducted, since the data were not normally distributed. The rest of the data from the three locations (Mannheim, Heidelberg, and Dettenheim) were analyzed separately using a generalized linear mixed model with negative binomial distribution (Figure 1 and Figure 2). Note that seasonal patterns were not statistically analyzed. Data from time point “0” were excluded from this analysis. The occurrence of aphids on individual sugar beet plants (yes or no) was analyzed separately for each location using a generalized mixed model with binomial distribution. Models were followed by Tukey tests [49]. The presence and number of natural enemies on the sugar beet could not be statistically analyzed due to the low number of individuals. Graphs were created using JMP Pro 17 (JMP Pro 17, 2023 JMP Statistical Discovery LLC, Cary, NC, USA).

## 3. Results

Only two aphid species, *M. persicae* and *A. fabae*, were found infesting the sugar beets. The numbers of natural enemies on the beet plants were too small for statistical analysis. The total numbers of natural enemies in Mannheim were as follows: hymenopteran parasitoids: 8; parasitized aphids: 4; coccinellidae (larvae and adults): 11; syrphid larvae: 1. Those for Heidelberg were as follows: hymenopteran parasitoids: 11; parasitized aphids: 21; coccinellidae (larvae and adults): 56; syrphid larvae: 2. In Dettenheim, the numbers were as follows: hymenopteran parasitoids: 17; parasitized aphids: 2; coccinellidae (larvae and adults): 21; syrphid larvae: 4 (for further information, please see Appendix A).

At the beginning of the experiment, before placing the dispensers in the field (i.e., at day 0), we did not observe any differences in the numbers of aphids between the two treatments at the different locations (Wilcox test, Mannheim: W = 79, *p* > 0.69; Heidelberg: W = 119, *p* = 0.79; Dettenheim: W = 127, *p* = 0.43) (Figure 1B,D,F). After we distributed dispensers containing FIMs across the experimental fields, we observed a significant reduction in aphid numbers in the FIM-treated patches in Mannheim compared with the control patches throughout the season (Figure 2A). This was also the case in Heidelberg, where aphid abundance was generally highest (Figure 2B), regardless of whether extreme outliers in the control treatments were included or excluded. No differences were found between the control and FIM-treated patches in Dettenheim (Figure 2C).

When analyzing the data in more detail, there was a general increase in the aphid population 14 d after application in Mannheim, followed by a decrease 28 d after application (Figure 1B). We documented significantly more aphids in the control patches than in the experimental patches with FIM dispensers at 14 d (*p* = 0.039 *), 28 d (*p* = 0.011 *), and 42 d (*p* = 0.0002 ***) after application, but not after 7 d, 21 d, or 35 d. In Heidelberg, we also observed an overall increase in aphid abundance starting at 14 days post-application in both the control and FIM-treated patches (Figure 1D). Twenty-eight days after application and at subsequent time points, the aphid populations had almost completely collapsed, regardless of treatment. As in Mannheim, we found significantly more aphids in the control than in the FIM-treated patches at 7 d (*p* < 0.000 ***), 14 d (*p* = 0.000 **), 28 (*p* = 0.015 *), 35 d (*p* = 0.010 *), and 42 d (*p* = 0.000) after application. In contrast to the other locations, we observed an increase in aphid abundance over the whole season in Dettenheim, and no differences between controls and FIM-treated patches (Figure 1F).

## 4. Discussion

So far, many studies have shown that the aphid alarm pheromone EBF repels aphids from colonizing plants, reduces aphid population growth, stimulates the production of alates (winged aphids) to leave their fully infested host plant [27], and attracts their natural enemies [50]. These effects indicate that EBF could be used for the control of aphid pests in crop plants. We studied the use of a mixture of commercial farnesene isomers (FIMs) containing synthetic EBF to control the aphid species *M. persicae* and *A. fabae* on sugar beet fields in three locations in Southern Germany.

While we did not find significant effects at one of the sites, at the two other sites, sugar beet plants in patches with FIM dispensers were found to be significantly less infected with aphids than those in control patches when the data for the entire season were analyzed together. Comparing experimental and control patches on the individual sampling dates, the number of aphids on the plants was reduced significantly in the FIM patches for at least half of the sample dates at these sites. This indicates that their is possible to achieve a reduction in aphid numbers by using FIM dispensers in sugar beet.

While it is unclear whether this reduction was due to a repellence effect on the aphids, a reduction in population growth, or the desertion of the host plants by alates, the effect of natural enemies can be excluded [48]. Their numbers were very low in general and did not increase in the presence of EBF. Possibly, the attraction of natural enemies with the use of EBF does not work in intensively managed agricultural landscapes with low biodiversity. Alternatively, it might well be that EBF does not act as a kairomone in aphid–natural enemy interaction under natural conditions and in natural concentrations. This was indicated by a detailed field study with pea aphids *Acyrthosiphon pisum* [36].

Generally, the observed reduction in the number of aphids in the presence of EBF was relatively small, and there was no effect on one of the locations. This agrees with earlier studies, which also found only small effects of EBF on aphid population size in crop fields of peas [51], wheat [51], cabbage [17], and iceberg lettuce [52]. Thus, EBF seems to have only a limited effect when studied under field conditions. Potential explanations for this reduced efficiency in the field in contrast to the laboratory include (1) the chemical instability of EBF, which oxidizes over time due to certain double bonds in the molecule, rendering the alarm pheromone ineffective [27,53], (2) the large distance between the points of release, (3) the low concentration of the EBF used in the experiments, and (4) the environmental conditions at different sites, which can lead to different microclimates and can make the FIM dispenser either more or less effective.

In addition, as field trials are difficult to standardize, there are certain aspects that could have influenced the numbers of aphids found on the plants. Firstly, the main wind direction was considered, but there is still the possibility of a change in wind direction. As a result, FIM could have been evaporated and delivered in the wrong direction, leaving the experimental patches unaffected, which may have affected aphid infestation. In addition, if the wind speed was too high or too low, the evaporated FIMs may have been carried either too far or not far enough to cover the entire expected area. Furthermore, due to the different responsibilities of the different farmers and the different weather and soil conditions at the different sites, sowing took place at different times (March to April), which could have influenced aphid infestation.

From these explanations, chemical instability is unlikely to have been involved in our experiment, because FIMs contain alpha-tocopherol as a stabilizer and we observed the effects of EBF in our experiments over several weeks. In contrast, the distance between the release points and the sampled plants could well have been too large (Mannheim: 21.1 m; Heidelberg: 36.1 m; Dettenheim: 17.7 m), especially in the absence of wind. However, small effects or no effects of EBF on aphids were also observed in studies in which the compound was released by the plants themselves [53,54], indicating that distance to the release point was not the only important factor. The EBF concentration necessary to induce a behavioral or physiological response by aphids is unclear. Dose–response studies are scarce and the amount of EBF required to stimulate a response in aphids has rarely been studied in terms of absolute concentrations in field experiments. For *M. persicae*, which we have also studied, van Emden et al. found identical behavioral responses to EBF when applied at 1, 10, and 100 µg mL^−1^ [55]. Jing-Gong et al. showed that the response of *M. persicae* was stronger towards 1 µg mL^−1^ EBF compared with 0.1 µg mL^−1^, although the responses of related aphids in the genus *Aphis* were mostly higher at lower concentrations [56]. In the present study, we applied 0.7 mL of an undiluted mixture containing 15.1% EBF in each dispenser, firstly, because we had carried out preliminary studies on the exposure of FIMs for seven days under laboratory conditions and found that the evaporation was low, and secondly, because we wanted to achieve long-term exposure under field conditions. Possibly, this concentration was too high to induce a response. More studies are required to identify the optimal field concentration of the mixture of farnesene isomers used.

## 5. Conclusions

In summary, our study revealed that aphid infestation in sugar beet fields can be reduced using a cost-efficient mixture of farnesene isomers (FIMs). Obviously, FIM dispensers could be an affordable alternative to insecticides. However, there is a strong need to understand the basics to optimize the current system and to increase the effect and make it more reliable. For instance, the optimal concentration of FIMs for field application is unclear. Further research is also required to elucidate the appropriate density of dispensers in the field, their optimal distance, and their position within the field, depending on the wind direction. Considering their high potential to control aphid populations in the field without harmful insecticides, more efforts to optimize the use of FIMs are certainly justified.

## Figures and Tables

**Figure 1 insects-15-00736-f001:**
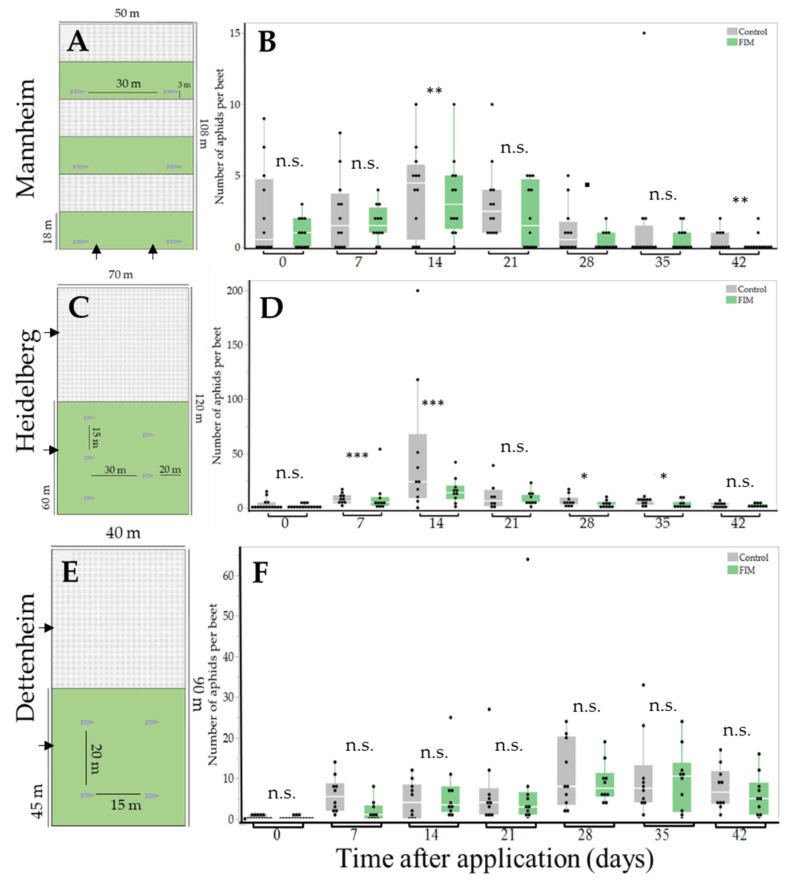
The use of FIM dispensers in sugar beet fields as a potential strategy to prevent aphid colonization and to inhibit colony growth. An overview of the experimental design (field patches with dispenser: green; without dispenser: gray) is shown on the left for Mannheim (**A**), Heidelberg (**C**), and Dettenheim (**E**) and the number of aphids found per sugar beet sampled are shown on the right for Mannheim (**B**), Heidelberg (**D**) and Dettenheim (**F**). Black arrows indicate typical wind direction. The graphs on the right show the number of aphids at specific sampling times over a period of 42 days. The data are presented as box–whisker plots. Boxes indicate interquartile ranges and whiskers indicate minimum and maximum data points. Black dots represent raw data, and white lines indicate median values. Note that for better visualization of the Heidelberg dataset, the outliers are excluded. The *y*-axis of each location is adjusted according to the number of individuals found, resulting in different *y*-axes for Mannheim, Heidelberg, and Dettenheim. “***” significant at *p* < 0.001, “**” significant at *p* < 0.01, “*” tendency with 0.05 > *p* < 0.06, “n.s.” not significant at *p* > 0.06.

**Figure 2 insects-15-00736-f002:**
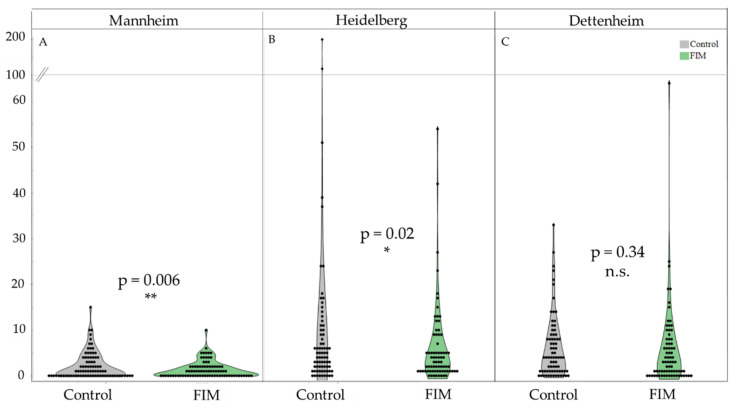
Total number of aphids found on *B. vulgaris* in the control patches (gray) and the patches with FIM dispensers (green) in Mannheim (**A**), Heidelberg (**B**), and Dettenheim (**C**). Data are presented as violin graphs to show the distribution of the raw data. Note that for better visualization of the Heidelberg dataset, the outliers are excluded. Furthermore, the *y*-axis has been interrupted to improve the visibility of individual data points. Dots represent raw data. “**” significant at *p* < 0.01, “*” significant at *p* < 0.05, “n.s.” not significant at *p* > 0.06.

## Data Availability

All data underlying our study have been deposited as Appendix A. There are no restrictions on data availability.

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
