# Peer review of "Can a Mixture of Farnesene Isomers Avert the Infestation of Aphids in Sugar Beet Crops?"

_insects, 2024, doi:10.3390/insects15100736_

Round 1

Reviewer 1 Report

Comments and Suggestions for Authors

In the manuscript of Kuhn et al., authors investigate the effects of EBF application on aphid infestation in sugar beet. The research is important and very timely, given the problems encountered in crop pest control programmes due to the development of resistance, coupled with the continuing limitation of available active substances. The manuscript is descriptive, the experiments well performed and the rationale easy to follow. I have just a few comments that, in my opinion, would improve the manuscript:

·       In the introduction, in the last sentence of the second paragraph «Results from field experiments on EBF as an attractant for natural enemies to control aphids, however, are inconsistent»,  I think you could give a little bit more information on this subject.

      In the Materials and Methods, in the first paragraph of the 2.4 you give a very good description about the experimental patches. Please clarify if the patches are separated from each other in some way, or not.

·     In the first paragraph of results, I understand that the number of natural enemies was too small for statistical analysis, but I think it would be reasonable to mention whether there has been any change/difference in their number(s).

·      In the fourth paragraph of the Discussion, among the potential explanations of the reduced efficiency in the field is the concentration of EBF, but personally I failed to understand how you determined the concentration you applied. Please clarify this. Additionally, I wonder if the environmental conditions and microclimate of each experimental area might play a role in this.

Reviewer 2 Report

Comments and Suggestions for Authors

Comments to the Author

This paper tested if the artificially release of the pheromone can be an environmentally friendly alternative to prevent aphid infestation in sugar beet. The results showed a reduction in aphid abundance in the FIM-treated patches in two out of three sites. Its indicate the potential of this pheromone as an aphid repellent in sugar beet production. However, its looked that the FIM-treated should be used in combination with other IPM measures. This is a very interesting topic.

The paper is well written.

I have some question:

1. why author did not counted the alates (winged aphids) number in the experiment?

2. The affects of FIMs to aphids and natural enemies are related to distance, so I want to know the distance between the sample of beet plants. I found the differences of the aphids and enemies among different days, Are you have the differences among different distances?

A reference “Bruce TJA, Aradottir GI, Smart LE, Martin JL, Caulfield JC, Doherty A, Sparks CA, Woodcock CM, Birkett MA, Napier JA, Jones HD. Pickett JA. 2015. The first crop plant genetically engineered to release an insect pheromone for defence. Scientific Reports, 5: 11183. DOI: https://doi.org/10.1038/srep11183.” maybe has help for authors.

Reviewer 3 Report

Comments and Suggestions for Authors

The authors conducted field studies at three different locations to explore the repellency effect of (E)-ß-farnesene (EBF) on two aphid species.

Field experiments always are difficult as the researchers have less control over extraneous/confounding variables that may influence the results. However, field experiments are valuable for understanding how findings from controlled laboratory studies can be applied to real-world situations. 

Introduction section

Please move the following paragraph to appropratite section of Materials and Methods 

"The mixture was used instead of pure EBF, because it is more cost effective. We applied dispensers filled with FIM on conventionally managed B. vulgaris fields in three locations in Southern Germany and monitored the number of aphids on the plants."

Materials and Methods 

The authors used different number of FIM dispensers (2 in Mannheim, 5 in Heidelberg, 4 in Dettenheim) and planting time (March-April 2023) and methods. These parameters might have had significant effects on the results? Why did not the authors conduct the experiments in equal conditions? Any specific reason for that?

The authors should clarify the following questions:

-how many plants were there in each plot? 

-The authors sampled different number of plants in Mannheim. Why?

-How did you chose the plant in rows? Randomly or did you use a different method? 

The authors could chose planting time (March-April 2023) and methods as a indenpendent factor. This can be the main reason behind the aphid abundance in Dettenheim.

Discussion

Wind direction-speed, the number of EBF dispensers and various Planting Methods might have had a significant influence on the results.

It might be a good idea to add a paragraph discussion section about these.

Please see the attachment for a detailed comments
